



# Application of quality-controlled sea level height observation at the central East China Sea: Assessment of sea level rise

Taek-bum Jeong[1,2], Yong Sun Kim[3,7], Hyeonsoo Cha[4], Kwang-Young Jeong[5], Mi-Jin Jang[3],

Jin-Yong Jeong[6], and Jae-Ho Lee[3*]

[1]Center for Climate Physics, Institute for Basic Science, Busan, Republic of Korea, 46241

[2]Pusan National University, Busan, Republic of Korea, 46241

[3]Ocean Circulation Research Center, Korea Institute of Ocean Science and Technology, Busan, Republic of Korea, 49111

[4]Center for Sea-Level Changes, Jeju National University, Jeju, Republic of Korea, 63243

[5]Ocean Research Division, Korea Hydrographic and Oceanographic Agency, Busan, Republic of Korea, 49111

[6]Marine Disaster Research Center, Korea Institute of Ocean Science and Technology, Busan, Republic of Korea, 49111

[7]Ocean Science, University of Science and Technology, Daejeon, Republic of Korea, 34113

*Correspondence to*: Jae-Ho Lee (Jaeholee@kiost.ac.kr)

**Abstract.** This study presents the state-of-the-art quality control (QC) process for sea level height (SLH) time series observed at the Ieodo Ocean Research Station (I-ROS) in the central East China Sea, a unique in-situ measurement in the open sea for over two decades with a 10-minute interval. The newly developed QC procedure called the Temporally And Locally Optimized Detection (TALOD) method has two notable differences in characteristics from the typical ones: 1) spatiotemporally optimized local range check based on the high-resolution tidal prediction model TPXO9, 2) considering the occurrence rate of a stuck value over a specific period. Besides, the TALOD adopts an extreme event flag (EEF) system to provide SLH characteristics during extreme weather. A comparison with the typical QC process, satellite altimetry, and reanalysis products demonstrates that the TALOD method can provide reliable SLH time series with few misclassifications. Through budget analysis, it was determined that the sea level rise at I-ORS is primarily caused by the barystatic effect, and the trend differences between observations, satellite, and physical processes are related to vertical land motion. It was confirmed through GNSS that ground subsidence of −0.89±0.47 mm/yr is occurring at I-ORS. As a representative of the East China Sea, this qualified SLH time series makes dynamics research possible spanning from a few hours of nonlinear waves to a decadal trend, along with simultaneously observed environmental variables from



the air-sea monitoring system in the research station. This TALOD QC method is designed for SLH observations
in the open ocean, but it can be generally applied to SLH data from tidal gauge stations in the coastal region.
**1 Introduction**
Sea Level Height (SLH) comprises oceanic components such as tides and currents and atmospheric components
(Pirooznia et al., 2016). Global warming due to the increased greenhouse gas has caused a persistent increase of
heat fluxes into the ocean, accelerating upper ocean heat content and the loss of land-based glaciers and ice sheets,
resulting in rapid sea level rise (SLR; Pugh, 2019; IPCC). This rise is not spatially homogeneous but localized in
association with a change in the current system (*e.g.,* Roemmich et al., 2007; Hamlington et al., 2020; Lee et al.,
2022; Li et al., 2024). Rising sea levels have induced coastal erosion and broad flooding, suggesting a presumable
vulnerability of populated low-lying coastal regions to global warming (Kulp and Strauss, 2019). Recent research
has demonstrated its robust relationship with extreme weather events (Cayan et al., 2008; Yin et al., 2020; Calafat
et al., 2022), underscoring the need for a long-term SLH monitoring network.
A global network of tidal gauges at the coastal region, along with satellite altimetry for the open ocean, has made
it possible to observe worldwide sea level changes (*e.g.,* Dieng et al., 2017; Cazenave et al., 2018; Chen et al.,
2017; Royston et al., 2020; Cha et al., 2023). The upward trend of global mean SLR increased from 3.05 mm/yr
for the period 1993–2018 to 3.59 mm/yr from 2006 to 2018, about twice faster than 1.7 mm/yr during the $20^{th}$
century (Fox-Kemper et al., 2021; Nerem et al., 2018). A future projected sea level trend is expected to be 4.63±1.1
mm/yr for the period 2010–2060 from observed and reconstructed measurements around Korea (Kim and Kim,
2017), implying more frequent occurrences of extreme weather and climate hazards associated with the mean sea
level rising within the near future.
Due to its broad socioeconomic implications, the Korea Hydrographic and Oceanographic Agency (KHOA) has
constructed a sea level monitoring network with thirty-eight tide gauge stations for the coastal region around
Korea (red pentagram in Fig 1). Besides, the ocean research stations, steel framed tower-type research facilities,
started to conduct unceasing and autonomous observations to cover a north-south section of the Yellow and East
China Seas, allowing us to understand air-sea interaction and atmospheric and oceanic processes in various time
scales at the open ocean (Ha et al., 2019; Kim et al., 2019; Kim et al., 2022; Kim et al., 2023a; Kim et al 2023b;
Saranya et al., 2024). The Ieodo ocean research station (I-ORS), the first one constructed at 32.125°N, 125.18°E
(see Fig. 1 for its location) in 2003, has produced sea level measurements using a radar-type sensor with a 10-
minute interval for more than two decades since October 2003. This station is strategically positioned along the



pathway of typhoons that impact the Korean Peninsula; hence, the I-ORS can serve as a crucial platform for
comprehending extreme weather phenomena (Moon et al., 2010; Park et al., 2019; Yang et al., 2022) and long-
term climate variability.
The collected sea level data, however, contains intricate outliers such as missing, spike, electric noise, stuck, drift,
systematic conversion (or offset)[1], and so on (Pytharouli et al., 2018). These outliers must be identified or
corrected before being used for research. This process, known as Quality Control (QC), involves outlier
classification into range, variability (or gradient), and sensor test categories (OOI, 2013; Min et al., 2020). Each
institution utilizes a different algorithm. For instance, outliers might be identified by applying a threshold of three
times the standard deviation above and below the average of measurements within a specified sliding window
(Min et al., 2020; 2021). This approach assumes the Gaussian distribution of the observed time series; hence, it
may not be suitable for uniformly applying this method because nonlinear waves or abrupt extreme events tend
to be misclassified as outliers. Also, the variables that are greatly affected by strong tides may have difficulty
detecting outliers when a range check is performed without considering tidal components. Therefore, Pugh (1987)
suggested a QC procedure based on tidal components estimated by a harmonic analysis. Recently, Pirooznia et al.
(2019) computed tides by adopting the classical least square (CLS) and total least square (TLS) from raw data
that contained outliers and missing values. They used the estimated tidal components to get residual components
of SLH data and then performed outlier detection. This process might be appropriate for the data stably obtained
from tide gauge stations but seems impertinent to measurements in the open ocean, which may have various types
of intricate outliers.
In addition, previous studies attempted to verify the factors contributing to sea level rise (SLR) using various data.
Cha et al. (2023) quantified and assessed the underlying processes contributing to sea level rise in the northwestern
Pacific using reanalysis data and satellite measurements from 1993 to 2017. This study found that the major
contributions to sea level rise are land ice melt and sterodynamic components, while the spatial pattern and
interannual variability are dominated by the sterodynamic effect. However, satellite-based sea level observations
cannot detect vertical land motion such as subsidence or uplift, which may lead to trend differences between

---

[1] The I-ORS methodology for sea level measurements was changed in December 2007. Previously, the I-ORS observed the length between the instrument and the sea level; since then, it has been changed to observe the sea level to the bottom. Due to the methodological switch, the recorded sea level time series has a sharp and systematic offset, as described in section 2.1.





satellite and station observation. This indicates the need to analyze the variability of vertical land motion at these
stations as well.
This paper aims to introduce a unique, invaluable SLH time series obtained in the open ocean over two decades,
processed with a newly developed QC process named the Temporally And Locally Optimized Detection (TALOD)
method. For this purpose, we take advantage of simulated tidal components based on TOPEX/Poseidon global
tidal model v9 (TPXO9; Erofeeva and Egbert, 2018). This high-resolution global tidal model reproduces tidal
well components around the Korean peninsula (Lee et al., 2022) and, hence, can be used for a local and temporal
range check. The performance of the newly suggested QC process is assessed by comparing it to a typical QC
method suggested by the Intergovernmental Oceanographic Commission (IOC), and the qualified, daily and
monthly averaged sea level time series are assessed using satellite altimetry and reanalyzed products from
GLORYS12, ORAS5, and HYCOM regarding their long-term trends. Additionally, the physical processes
contributing to sea level rise at the I-ORS were analyzed using reanalyzed product, and the vertical land motion
at the I-ORS platform was estimated using the Global Navigation Satellite System (GNSS).

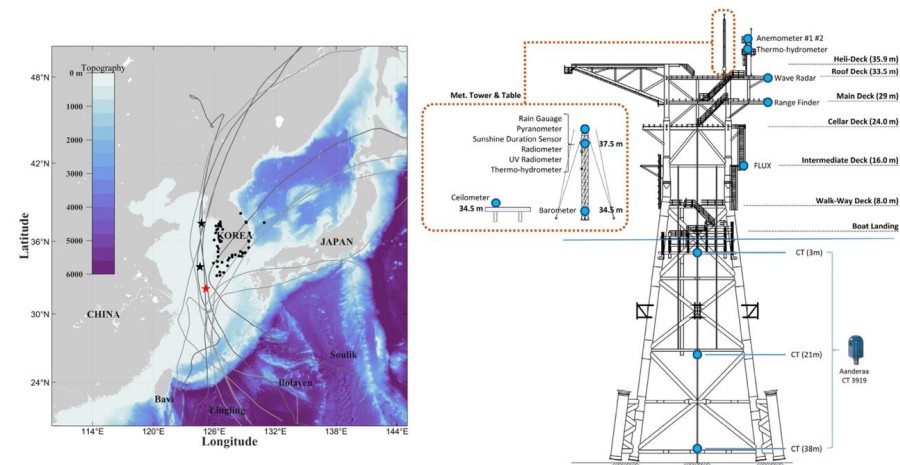


**Figure 1. The structure of I-ORS and Instruments (Right) and the horizontal distribution for bathymetry and the**
**tracks of typhoon passed by I-ORS (data from Joint Typhoon Warning Center; cases depicted in Fig. 10). The star**
**marks indicate the location of the I-ROS (red) and the Socheongcho (black; above) and Gageocho (black; below) Ocean**
**Research Station, respectively. The black dots depict the locations of tide stations. The grey solid lines show the storm**
**tracks passing by I-ROS from 2003 to 2022. The darker lines indicate the typhoon case in Table 2.**



**2 Data and Method**

**2.1 SLH observed time series from the I-ORS**

We constructed the TALOD QC process based on the TPXO9 and applied it to the 10-minute interval real-time SLH measurements obtained from the I-ORS, a total of 1,011,584 data points from 8 October 2003 to 31 December 2022. The data was measured by the MIROS SM-140 non-directional wave radar, installed at the main deck 29 m above the sea surface (Fig. 1). The range finder principally estimates the distance to the sea surface through the reflected signals by detecting back-scattered microwaves from the surface. Table 1 describes the detailed specification of the SM-140. The sensor's measurements are known to be relatively free from atmospheric conditions such as rain, fog, and water spray.

As mentioned in the introduction, the sea level measuring standard was changed on 12 December 2007. A sharp offset of about 6.7 m, therefore, was recorded between the data before and after the transition point (TP; see Fig. 2). Before the TP, the range finder recorded the distance from the sensor to the sea surface as sea level. After that, the KHOA altered the standard to record the actual sea level by subtracting the measured distance from the known height from the sea bottom to the sensor (KHOA, 2013). Therefore, this study corrected the forepart by flipping it upside down and then shifting to the position extrapolated to the first time of the data afterward. Also, we performed the harmonic analysis on the corrected SLH time series to validate the correction method. The corrected SLH time series for December 2007 estimated a sufficiently high signal-to-noise ratio (SNR) over 10.0 (Pawlowicz et al., 2002), compared to the much broader ranges like years or decades of SLH at I-ORS. Its consistencies in amplitude and phase with the rear subset also guaranteed the method for correcting the systematic offset.

Table 1. Instrument specifications for the SM-140 by MIROS.

| Data | Range | Resolution | Accuracy |
|---|---|---|---|
| **Range** | 1 – 23 m | 1 mm | < 5 mm |
| | 3 – 95 m | | |
| **Frequency** | 50 – 200 Hz (according to range) | | |



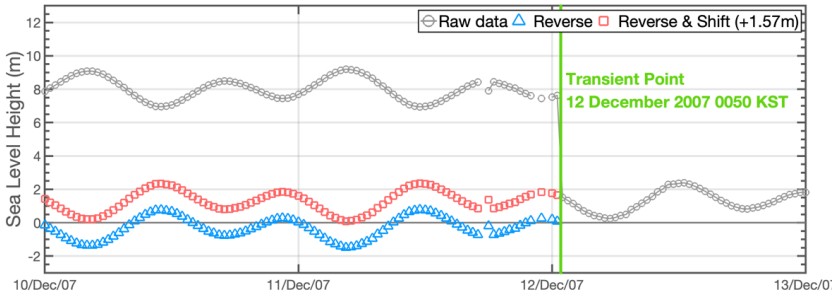

126

**Figure 2. The circle markers indicate each process of methodological adjustment for the data before TP. The grey line with circles means the raw data and blue and red marker lines indicate the reverse and shift (+ 1.57m after reversed) process.**

**2.1.1 Satellite altimetry and reanalysis products**

We collected satellite altimetry and reanalysis datasets to validate the performance of the qualified SLH. The satellite is the gridded L4 sea surface height dataset provided by Copernicus Marine Environment Monitoring Service (CMEMS, https://doi.org/10.48670/moi-00145) for 1993-2022. This altimetry, sea surface height from the geoid, was calculated through optimal interpolation (OI) by merging along-track altimetry from all satellites. Inverted barometric and tidal heights correction was applied to adjust the along-track data. The daily gridded satellite altimetry has a quarter-degree resolution for the global ocean. We used daily SSH time series at the nearest grid point to the I-ORS.

The three SSH products used in this study are the HYbrid Coordinate Ocean Model (HYCOM, https://www.hycom.org/) data-assimilative reanalysis (HYCOM-R) for the period of 2003-2017 and HYCOM non-assimilative simulation (HYCOM-S) from 2018-2022, Global Ocean Physics Reanalysis 12 version 1 (hereafter GLORYS; Lellouche et al., 2021), and the Ocean Reanalysis System 5 (hereafter ORAS5; Zuo et al., 2019). The HYCOM product provided by the Navy's operational Altimeter Processing System (ALPS) has a spatial resolution of 1/12° by 1/12° for the global ocean and a temporal resolution of 3 hourly. GLORYS12 is produced by Mercator Ocean International (https://www.mercator-ocean.fr/en/) and has a spatial resolution of 1/12° by 1/12° for the global ocean with a daily resolution. The ORAS5 provided by the European Center for Medium-Range Weather Forecasts (ECMWF) has a spatial resolution of 1/4° by 1/4° for the global ocean and a temporal resolution of monthly (DOI: 10.24381/cds.67e8eeb7). To efficiently compare sea level variability, the SLH of all datasets was converted to sea level anomalies by subtracting their mean values. Except for ORAS5, which is monthly data, the other sea level data were averaged daily. Similarly, we estimated the daily mean observed time series when more than half of the data were available or flagged as good data.



## 2.2 TALOD QC

### 2.2.1 Meta check

After correcting the systematic offset in the observed sea level time series, we classified outliers into four

categories: metadata, range, spike, and stuck (see Fig. 3 for a flowchart). The metadata check involves manually

flagging unreliable data, including instrumental jolts or a data section that may disrupt the following automatic

detection procedures to prevent contamination of the observed data's long-term characteristics. This examination

is normally based on historical metadata information (or field notes) on the sensor's maintenance, cleansing, a

power shortage event in the ocean research station, etc. Unfortunately, the observed SLH time series from the I-

ORS are not distributed with metadata information. Instead, we flagged subjectively a section where the

periodicity of SLH data was irregular or nonsensical data existed for several days. For example, from June 2016

to July 2017, the sea level observations at the I-ORS involved two relocations and one replacement of the

observational instrument, and the sea levels observed during this period were relatively low (not shown). As a

result, 56,024 data points were flagged based on the metadata check accounting for 6.32% of the total observations.

This study points out the need for recorded metadata information to ensure quality assessment of the observed

time series and efficient instrumental maintenance.

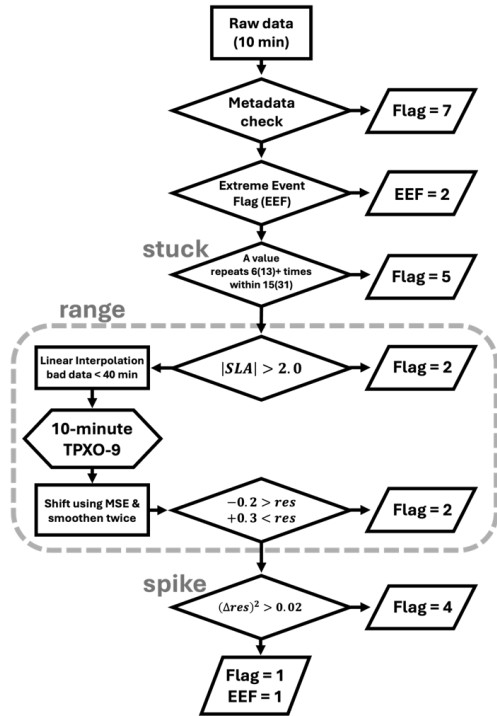





**Figure 3. Flow chart of TALOD QC process.**

**2.2.2 Stuck check**

After the metadata check, we recommend examining stuck values in the time series. Generally, a stuck check detects outliers when a fixed value is continuously recorded over a certain period. At the I-ORS, the SLH measurements exhibit two distinct characteristics of stuck values. Firstly, these values persist for a certain duration without variation; a typical QC process can identify this kind of stuck. An abnormal case is observed at the I-ORS: alternation between normal observations (good data) and fixed values. To handle this unusual stuck case efficiently, we adopted the density of identical values over a certain period. We experimented with various range and frequency combinations. As a result, we flagged the cases when a single value was detected more than 6 times within a range of 15 or more than 13 times within a range of 31.

**2.2.3 Range check**

Normally, range check can be divided into two parts. A local or gross range check designates a single value that is difficult to occur naturally for a target variable at a specific location during the monitoring span. And seasonally varying range check effectively detects errors for variables dominated by seasonal variability, such as air or sea surface temperatures or humidity. However, these methods are not suitable for SLH measurements in shallow water with large tidal amplitudes, such as the maximum tidal amplitude of 2.5 m that can occur at the I-ORS, and significant seasonal cycles (Lee et al., 2006).

This study's range check consists of two procedures: a gross range check with a fixed range by assigning upper (+2.0 m) and lower (–2.0 m) limits for SLA, and a localized check with temporally varying ranges by taking advantage of the tidal prediction model. The gross range check effectively identifies extremely high values such as 29.0 m and 7.98 m, which are frequently recorded in the SLH measurements from the I-ORS even during normal situations. For the local range check, we used the TPXO9 tidal model, which has a 1/30° horizontal resolution. This global tide model offers realistic spatial and temporal tides around the Korean Peninsula with the smallest root mean square difference (RMSD) compared to tide gauge observations (Lee et al., 2022).

Tide data extracted from the TPXO9 sliding every month was adjusted using the observed SLH for the same period (Fig. 4). A month window is selected to consider seasonal evolution. The extracted tidal time series was shifted to positions where the Root Mean Square Errors (RMSEs) are minimized (the red line in Fig. 4). Overshooting tends to be generated when using the arithmetic mean only for the shifting, especially for the convex-up and -down data, which correspond to high and low tides respectively, thus potentially resulting in



196    detecting overestimated outliers. To address the overshooting issue, the residual time series, i.e., the observations

197    minus mean shifted tides, is smoothed twice and then added to the estimated tidal time series (the green line in

198    Fig. 4). When the difference between the observed SLH and the bias-corrected tide exceeds +0.3 meters or falls

199    below –0.2 meters, the local range check identifies it as an outlier (see Fig. 5b). These thresholds are sufficient

200    for elevation changes associated with nonlinear internal waves in this region (Lee et al., 2006).

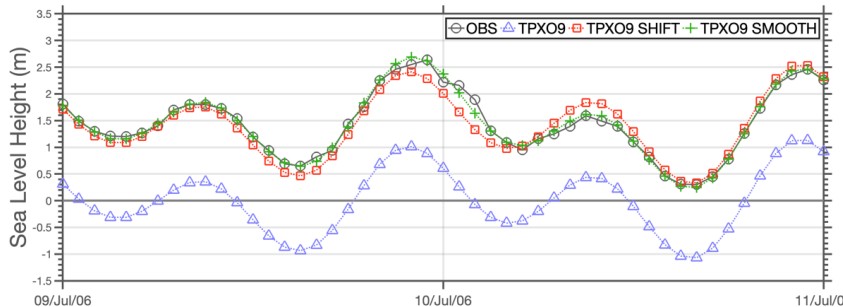

201

**Figure 4. Lines indicate the processes for fitting TPXO9 to observation (black line with circle) in the range check. (1)**
**The blue line with a triangle means raw TPXO9 data. (2) The orange line with the square shows mean-shifted TPXO9**
**based on the Mean Square Error method. (3) The green line with a circle indicates the final output with a twice-**
**smoothened bias added.**

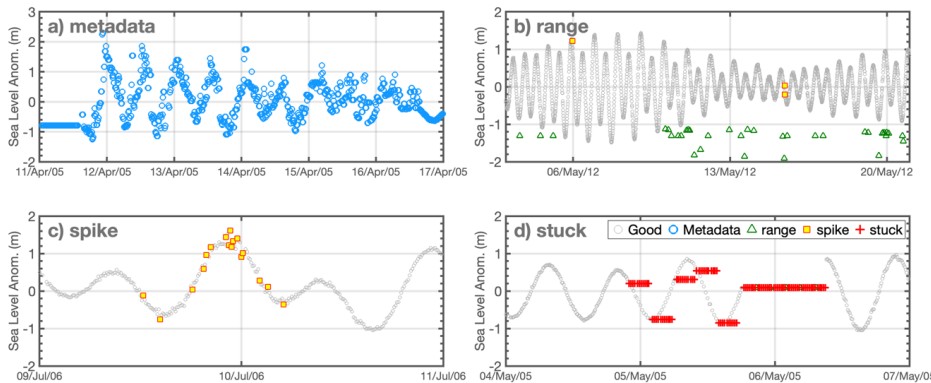

206

**Figure 5. Time series for the examples of 4 flags. a) metadata, b) range, c) spike, and d) stuck. Each marker indicates**
**Good Data (grey circle), metadata (blue circle), range (green triangle), spike (yellow square with red outline), and stuck**
**(red cross), respectively.**

**2.2.4 Spike check**

The spike check is developed based on the Gradient Spike Method (GSM) following Hwang et al. (2022). The

GSM generally detects outliers using the gradient of SLH data. However, we employed the temporal discrepancy

in the non-tidal residual SLH time series; that is, if the square of that value exceeds 0.02, it is classified as a spike.

The equation is as follows:



$$flag = find((\Delta residual)^2 > 0.02),  \tag{1}$$


**2.2.5 Extreme event flag**
Atmospheric factors such as sea level pressure and wind modulate SLH; the inverted barometer effect (IBE) and
strong winds can generate abrupt SLH fluctuations. Under extreme weather, the SLH measurements can be
classified as an outlier through range and spike checks. The flagged SLH data during severe weather might be
regarded as good data, depending on the situation. As a last QC procedure, this study introduced the extreme event
flag (EEF) to note that the SLH data was measured over severe weather periods. The typhoon cases analyzed in
this study are shown in Table 2.
The observed range of sea surface height anomalies was almost equal for both normal and typhoon situations, i.e.,
0.30/–0.20 m and 0.29/–0.20 m, respectively. However, there was a significant difference in variance, which
implies large fluctuations in the SLH measurements. The normal case exhibited a variance of 9.0 cm$^2$, whereas
during the typhoon-influenced period, it increased to 40 cm$^2$, approximately five times higher. Consequently,
although the maximum/minimum ranges of residual components remained almost unchanged during typhoon
periods, the outliers classified by the spikes increased significantly (Fig. 6). We manually flagged the typhoon
period with the EEF based on the daily variance and reported information on typhoons from the KMA.
**Table 2. List of Typhoon cases during observation.**

| Typhoon | Start date | End date |
|---|---|---|
| **Chanthu (2021)** | 14 Sep, 2021 | 16 Sep, 2021 |
| **Bavi (2020)** | 26 Aug, 2020 | 26 Aug, 2020 |
| **Lingling (2019)** | 6 Sep, 2019 | 7 Sep, 2019 |
| **Kong-rey (2018)** | 6 Sep, 2018 | 7 Sep, 2018 |
| **Soulik (2018)** | 22 Aug, 2018 | 23 Aug, 2018 |
| **Chan-hom (2015)** | 12 Jul, 2015 | 12 Jul, 2015 |
| **Neoguri (2014)** | 9 Aug, 2014 | 9 Aug, 2014 |
| **Bolaven (2012)** | 27 Aug, 2012 | 28 Aug, 2012 |
| **Muifa (2011)** | 8 Aug, 2011 | 9 Aug, 2011 |
| **Megi (2004)** | 10 Aug, 2004 | 10 Aug, 2004 |



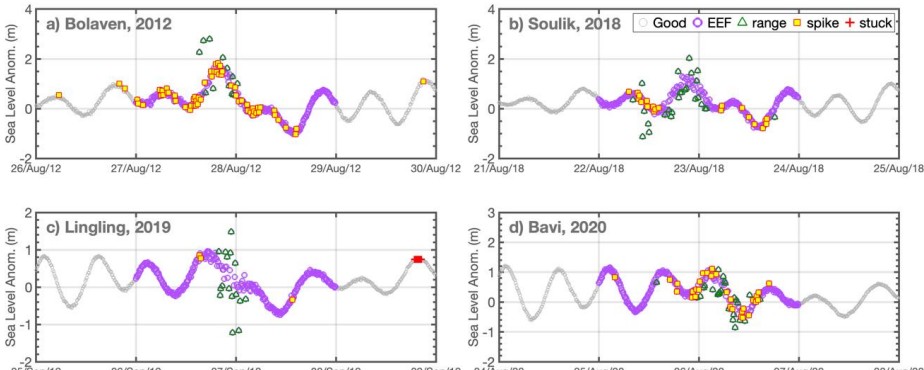

**Figure 6. Same as Fig. 5, but for Typhoon cases.**

**3 results**

**3.1 Comparative analysis to existing QC process**

Representative results obtained during the TALOD QC are shown in Figure 7, and the number of outliers and

proportions flagged by each QC process are presented in Table 3. The results were compared with those obtained

by applying the IOC's standard QC process to assess the performance of the TALOD QC process. The IOC was

designed and applied as a QC procedure consisting of several steps to accord with international standards through

the support of the National Data Buoy Center (NDBC) and the National Science Foundation under the National

Oceanic and Atmospheric Administration (NOAA) to provide uniformly qualified observations to scientists (Min

et al., 2020). The differences between those two QC processes are illustrated in Figure 8 and summarized in Table

4.

We collected a total of 1,011,584 SLH data observed at I-ORS during the observation period from 2003 to 2022.

After excluding 165,702 instances of missing values (NaNs), 886,128 data points were kept for quality control

and analysis. Of these, 793,034 (89.49%) were classified as good data, while 93,184 data points (10.51%) were

flagged as bad through the TALOD QC procedure (Table 3). Among the flagged data, excluding those flagged as

the meta, stuck values constituted the majority, representing 89.84% of the bad data. This was followed by spike

and range flags, accounting for 5.52% and 4.64% of the bad data, respectively.

Seasonal patterns in the frequency of each flag were further analyzed. The number of occurrences of bad data was

found to be the highest in spring, exceeding the annual average by a factor of 1.28. This seasonal increase was

primarily driven by a higher occurrence rate of stuck errors. Specifically, a total of 33,383 stuck errors were



recorded, with 16,536 instances occurring in spring, the highest count across all seasons (winter: 5,795; summer:
7,985; autumn: 3,067). The spring frequency of stuck errors was nearly double the annual average (1.98 times).
Other bad data types, such as range and spike, exhibited relatively low frequencies throughout the whole season,
with total counts of 1,725 and 2,052, respectively. Conversely, the meta-flagged data, which accounted for the
largest proportion of bad data excluding NaN values, displayed a uniform distribution across all seasons, with a
mean of 56,024 occurrences (winter: 14,934; spring: 12,298; summer: 14,843; autumn: 13,949). As a result, the
meta flag did not contribute significantly to the observed seasonal variations in the long-term perspective.
The overshooting-like errors related to extreme weather conditions, such as range and spike flags showed peak
occurrence rates in summer. This seasonal pattern coincided with the peak typhoon season over the NWP,
indicating a linkage between extreme weather events and the occurrence of overshooting-like error types.
The SLH is dominated by neap-spring tidal cycles, and it can induce misclassifications in error detection by a
range check that adopts a constant value as a threshold. However, the TALOD method utilizes residual
components that consider the rapid increase/decrease of SLH caused by most diurnal components and short-
duration weather systems, thereby reducing detection errors. For example, the range check in the TALOD QC
process successfully flagged 1,936 data points by outliers. In detail, the gross range check detected 1,121 bad data,
while the temporal and local outlier detection identified 815 instances of bad data. As a result, the temporally and
locally utilized outlier detection method successfully captured the errors with little biases. The TALOD QC
process preemptively flags bad data that excessively disrupt continuity through the range checks. This approach,
as depicted in Figure 8f, prevents detection failures caused by recurrent spike error values. The IOC's spike check
has trouble with flagging spike-type errors within a short period. These unqualified outlying values may provoke
the downgrading in the performance of the spike check using min/max for calculating threshold. Attention should
be given when applying the IOC QC processes to such sea level measurements because the automatic QC on
observation data could be vulnerable to recurrently recorded spike-like errors. For instance, among the 261
observations logged from 1 June 2016 00 KST to 14 June 2016 00 KST, the TALOD method flagged 43 instances
as bad data, while IOC identified 37 values only with apparent error-like values still remaining (see Fig. 8e and
8f).
Moreover, as summarized in Table 4, the two QC processes showed significant differences in the stuck check.
While the TALOD QC process successfully detects stuck values, as illustrated in Figure 8a, 8c, 8e, and 8g, the
IOC seems to fail to identify these error-like values. Instead of flagging abnormal stuck values, the IOC QC
removes the entire section (Fig. 8b, 8d, 8f, and 8h). Furthermore, the IOC's stuck check, which is designed to



identify values as stuck when the sensor records the same values, tends to classify excessively normal data into
stuck errors due to instrumental issues including low frequency (10 minutes); these situations are frequently
observed during high and leap tides (Fig. 8d).
During the application of the IOC Process to SLH data, misclassifications or detection failures were confirmed
due to the inability to identify irregularly repeated stuck errors. However, the TALOD applied optimized detection
techniques, and 45,850 stuck errors were successfully flagged. Figure 9 shows the distribution of observed and
qualified SLAs. Compared to the idealized normal distribution indicated by the grey line in Figure 9, unusually
high values were concentrated in the ranges of –1.4 to –1.3 m, –0.2 to –0.1 m, and 0.4 to 0.5 m. After the TALOD
QC, this distribution is more closely aligned with the normal distribution, indirectly suggesting the performance
of the TALOD QC to identify outliers.
**Table 3. Detection counts and proportions for each flag from Oct 2003 to Dec 2022 (excluding NaN values).**

| Flag number | 1 | 2 | 4 | 5 | 7 | 8 |
|---|---|---|---|---|---|---|
| (Name) | (Good data) | (Range) | (Spike) | (Stuck) | (Metadata) | (NaN) |
| # | 793,034 | 1,725 | 2,052 | 33,383 | 56,024 | 165,702 |
| % (without NaN) | 89.49% | 0.19% | 0.23% | 3.77% | 6.32% | |


**Table 4:** The differences in flag detection methods between TALOD and IOC.

| Flag | TALOD | IOC |
|---|---|---|
| **Range** | Data point where observation exceeds the threshold from the tidal component, which is adjusted according to temporal observations | Data point exceeds sensor or operator-selected min/max for whole period |
| **SPIKE** | Data point where the square of the difference in residuals exceeds the threshold | Data point n-1 exceeds a selected threshold relative to adjacent data points |
| **STUCK** | Data point where the reoccurrence rates for constant value within the windows are over thresholds | Invariant value |





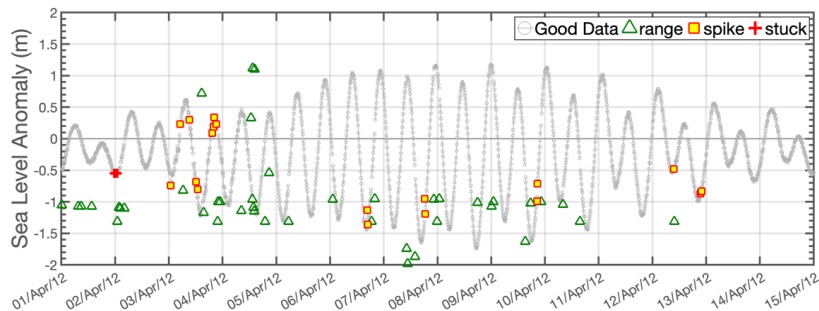

**Figure 7. Representative results from 01 Apr 2012 to 15 Apr 2012**

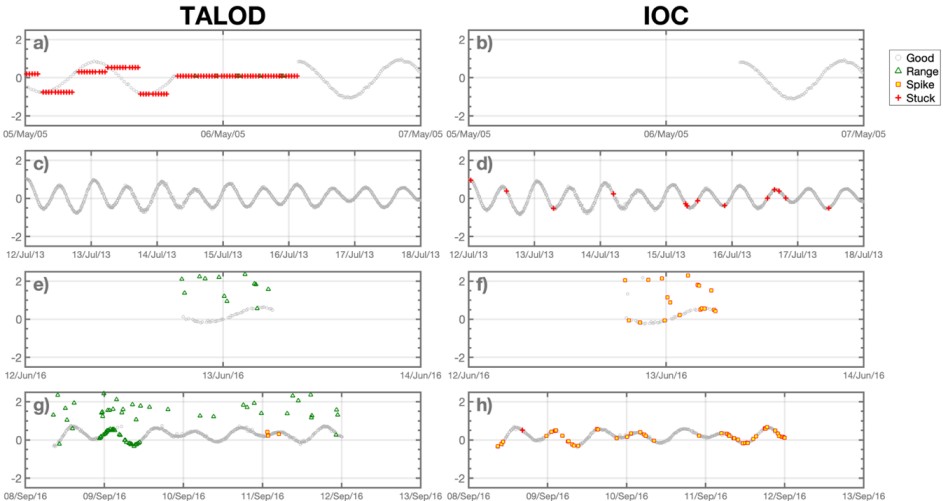

**Figure 8. Same as Fig. 5, but for invariant stuck case (a-b, from 05 May 2005 to 07 May 2005), stuck case during short-period (c-d, from 12 Jul 2013 to 18 Jul 2013), range-spike misclassification case (e-f, from 12 Jun 2016 to 14 Jun 2016), and range-spike mixed case (g-h, 08 Sep 2016 to 13 Sep 2016). The figures on the left and right sides show results for TALOD and IOC, respectively.**

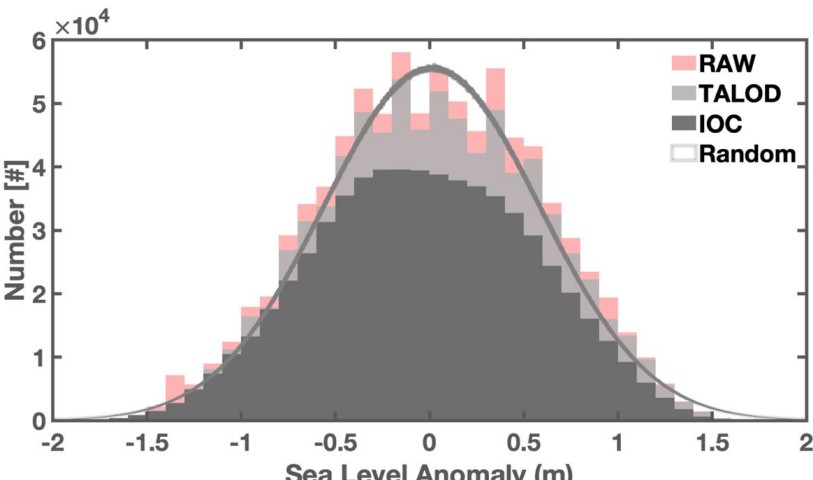

302

**Figure 9. Histogram of observed sea level anomaly without QC (light red) and with QC (light grey) from 2003 to 2022**
**at I-ORS. the area enclosed by a darker grey line indicates the normal distribution.**

**3.2 Data validation by using observation data**

Figure 10 displays the daily time series of SLA for each dataset except ORAS5. SLH generally represents the

vertically integrated heat contents of the ocean. Therefore, there are higher SLAs during the boreal summer period,

June-September, and lower SLAs during the boreal winter, December-March. The daily mean sea level range is

approximately ± 0.6 m for the observed one, −0.4 to +0.6 for the HYCOM product, and ± 0.3 m for GLORYS and

satellite altimetry. We calculated the standard deviation (STD) and variance of each dataset to infer their

variability and distribution. The STD and variance for the I-ORS measurements were 0.16 m and 0.02 m,

respectively. For satellite and GLORYS, the values were the same at 0.10 m and 0.01 m. The HYCOM-R had

values of 0.11m and 0.01m. Both Satellite and the two reanalysis data simulated lower variability of SLH

compared to the in-situ observation. However, both datasets captured the overall pattern well, showing high

accuracy with a low RMSE of less than 0.1m. Compared to HYCOM, which has a spatial resolution of 1/12° and

a temporal resolution of 3-hourly, the satellite exhibits lower seasonal variance, which might be due to substantial

optimal interpolation procedure to reduce high-frequency noise during a gridding process. Besides, significant

statistical differences were found between HYCOM and other datasets (OBS and reanalysis data) for the period

after 2018. Therefore, we further analyzed the HYCOM data by dividing it into two periods: before 2018

(HYCOM-R) and after 2018 (HYCOM-S).

First, we compared the SLR rates of each dataset (Fig. 10). The observation exhibited a SLR of 5.27 mm/yr for

this period from 2003 to 2022, while the satellite altimetry rendered slightly lower rates of 2.76 mm/yr. Owing to



a robust falling trend in the HYCOM's SLA during the recent period since 2018 (–24.42 mm/yr; HYCOM-S), the
overall rate of SLR for the HYCOM was negative (–4.22 mm/yr) during the study period, but the HYCOM-R has
a 2.70 mm/yr trend from 2003 to 2017. This result might indicate that we must be careful when using the HYCOM-
R and HYCOM-S products to study long-term climate dynamics.
Figure 11a shows the monthly sea level trends for the observation and other four datasets. The observation showed
a higher sea level rise rate (5.27±0.46 mm/yr) compared to the other datasets. ORAS5 exhibited a trend similar to
satellite altimetry, while GLORYS and HYCOM showed a sea level fall trend. As mentioned earlier, HYCOM
showed a strong fall trend unlike other datasets because it simulated lower sea levels after 2018. Also, we
compared the correlation and variability between the observation and the other four datasets using a Taylor
Diagram (Fig. 11b). Satellite altimetry exhibited the highest accuracy among the datasets, with a high correlation
coefficient (0.71) and low RMSE (0.04 m) compared to the observation. For HYCOM, it showed the lowest
correlation coefficient (-0.08) and highest RMSE (0.10 m) over the entire period, indicating poor agreement.
HYCOM-R demonstrated performance close to Satellite, whereas HYCOM-S exhibited a significantly low
correlation coefficient (-0.39) and high RMSE (0.12 m). The correlation coefficients of ORAS5 and GLORYS
were 0.71 and 0.76, respectively, and the RMSE of both data was 0.1 m, showing higher correlation and accuracy
than HYCOM. HYCOM was found to have an overall lower performance due to its inability to simulate the
variability of SLH since 2018 in HYCOM-S.

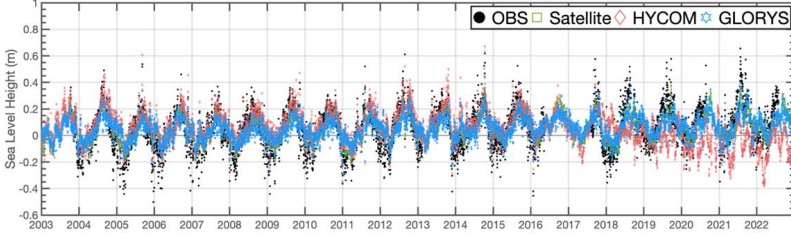


**Figure 10. Time series of monthly QC-ed observations (black dot), Satellite (green empty circle), HYCOM (light red**
**diamond), and GlORYS12 (light cyan hexagram) data during the observation period at the I-ORS.**





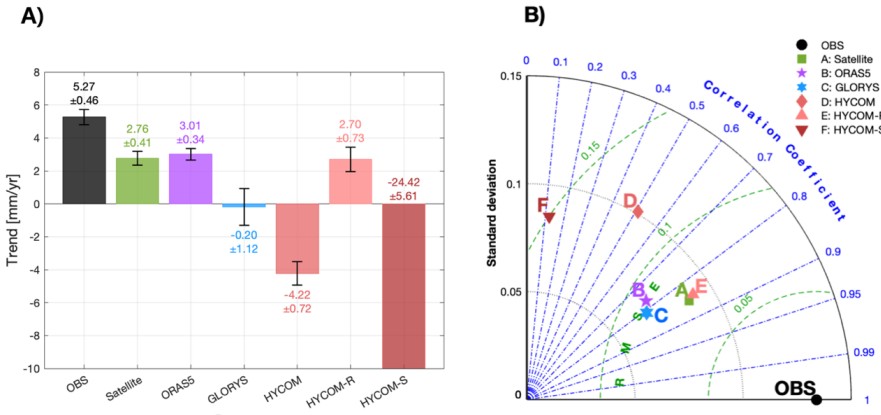


**Figure 11. Bar plot with error bar (A; Left) and Modified Taylor diagram (B; Right). the azimuthal angle represents**
**the correlation coefficient, the radial distance indicates the standard deviation, and the semicircles centered at the**
**"OBS" marker mean the Root Mean Square Errors. The colors and markers indicate each data (black circle:**
**observation, green square: Satellite, light cyan hexagram: GLORYS, purple pentagram: ORAS5, red diamond:**
**HYCOM, light red upward-pointing triangle: HYCOM-R, light red downward-pointing triangle: HYCOM-S).**

**3.3 Sea-level budget assessment at I-ORS**

As mentioned above, the SLH of the I-ORS produced through the developed QC process estimated a SLR rate of

5.27±0.46 mm/yr. Sea level change is divided into relative and geocentric sea level change representing the

distance from the sea floor and center of the earth to the sea surface, respectively. The ground-based observations

such as I-ORS are relative sea level. and its change can be affected by various physical processes including sea

level change due to ocean density and circulation (sterodynamic effect), mass exchange between the ocean and

land (barystatic effect), glacial isostatic adjustment (GIA) (Gregory et al., 2019; Frederikse et al., 2020; Cha et al.,

2024). In this regard, we performed a budget analysis of each physical process affecting SLR at the I-ORS.

The sterodynamic (SD) effect is calculated as the sum of dynamic sea level change (DSL) and global mean steric

sea level rise (GMSSL) (Gregory et al., 2019). DSL was obtained from ORAS5, which was also used for validation

data in this study. GMSSL used in-situ observation data provided by the Institute of Atmospheric Physics (IAP,

Cheng et al., 2017), Met Office Hadley Centre (EN4, Good et al., 2013), and Japan Meteorological Agency (JMA,

Ishii et al., 2017). GMSSL was produced using the temperature-salinity profile data from each institution and was

used to compute the SD effect by adding the DSL. The barystatic (BS) effect is the sum of ice melting from the

Antarctica, and Greenland ice sheets, glaciers, and changes in land water storage. Here, we used ocean mass

reconstructed barystatic data from Ludwigsen et al. (2024). GIA comprises sea level changes due to the

disappearance of glaciers since the glacial period, and we took the model results from Caron et al. (2018). Caron

et al. (2018) utilized a global positioning system (GPS) time series from 459 sites and 11,451 relative sea level



data to improve the model accuracy, and based on this, computed the ensemble mean of 128,000 model simulation
results.
Figure 12 shows the sea level time series and trend budget at the I-ORS along with a comparison to satellite
altimetry. The sea level change rate due to physical processes (Sum=SD+BS+GIA) was 2.57±0.35 mm/yr, about
2.70±0.58 smaller than the observation (5.27±0.46 mm/yr). This discrepancy was also found in comparing satellite
altimetry and observation (diff: 2.51±0.62 mm/yr). Among the components for physical processes, SD contributed
0.73±0.34 mm/yr, approximately 28% of the rise. The BS effect had the largest contribution, at 1.85±0.02 mm/yr
(about 72%). Meanwhile, GIA led to a slight fall in sea level, contributing -0.11±0.00 mm/yr, about 0.04%.
Satellites cannot detect vertical land motion (VLM) because they measure the change in distance from the center
of the earth to the sea surface, whereas station observations such as I-ORS are affected by VLM because they
measure the change in height from the sea floor to sea level (Han et al., 2014; Gregory et al., 2019; Cha et al.,
2024).Thus the difference between the sea level trend from satellite altimetry and I-ORS can be regarded as VLM
component, we checked whether a difference of approximately 2.51±0.62 mm/yr was associated with VLM. Cha
et al. (2024) defined the total VLM as the sum of the VLM components in GIA, BS, and local processes, where
GIA and BS are categorized as natural processes. The VLM of GIA was obtained from Caron et al. (2018), the
VLM of BS used the data of Frederikse et al. (2020), and the VLM component of the local process was calculated
using the difference between sea level change due to physical processes (2.57±0.35 mm/yr) and sea level change
from observation (5.27±0.46 mm/yr). At the I-ORS location, the VLM of GIA was calculated to be 0.22±0.14
mm/yr, the VLM of BS was 0.28±0.64 mm/yr, and the VLM of the local process was –2.67±0.60 mm/yr.
Therefore, the total VLM was approximately –2.17±0.89 mm/yr, indicating significant ground subsidence at the
I-ORS location, and this subsidence was more affected by local processes than by natural effects such as GIA and
BS.
Additionally, we analyzed the trend of observed vertical displacements using the Global Navigation Satellite
System (GNSS) observing 30-second intervals at the I-ORS from 2013 to 2019. The trend of GNSS vertical
displacements was –0.89±0.47 mm/yr, using daily mean. it's smaller than the VLM of the local process (2.67±0.60
mm/yr), but it certified that the actual ground subsidence exists.



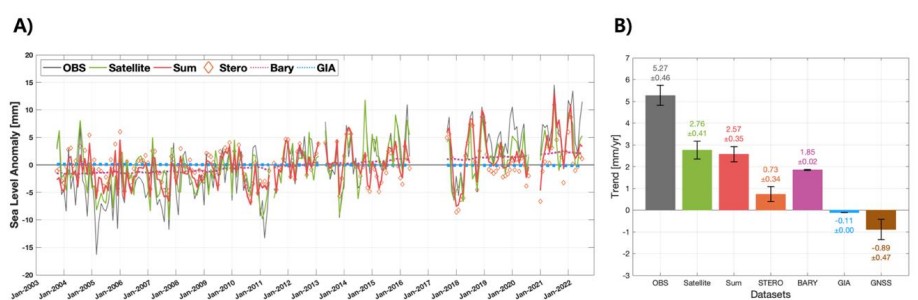


**Figure 12. Monthly time series of sea level anomalies (left) and bar chart with error bar for sea level rise rate (right; units: mm/yr). Each color and type of line indicates the dataset (OBS: black solid line, Satellite: green solid line, Sum: bright red solid line, STERO: orange diamond, BARY: purple dotted line, GIA: sky-blue dotted line, and GNSS: bright brown).**

## 4 Summary and Discussion

This study developed a novel quality control procedure based on a high-resolution tidal prediction model, named the Temporally and Locally Optimized Detection (TALOD) method, and applied it to 10-minute interval real-time SLH data observed by the MIROS Range Finder (SM-140) from 2003 to 2022. The TALOD method is divided into manual and automatic processes. The manual process includes a METADATA check that relies on the empirical knowledge of the data producer. The METADATA check flags sections that could contaminate the long-term characteristics of the collected time series observations. This check improves the performance of subsequent automatic QC processes. The automatic process includes RANGE, SPIKE, and STUCK checks. The range check with residual components derived from the tidal prediction model, TPXO9, may enable it to address known issues such as detection failure due to non-periodic outliers or adulteration when estimating the tidal components using the least square method. Spatiotemporally optimized thresholds reduce misclassification and detection failures caused by frequent error values during the spike check. The spike check detected bad data by setting a spatially and temporally optimized threshold using the non-tidal residual component. This approach can reduce false detections compared to the gradient-based GSM. Also, the GSM method tends to detect rapidly fluctuating SLH, such as extreme weather events, as an outlier. In the stuck check, we also utilized the occurrence frequency of specific values to handle the alternating of the good and bad data, the unique characteristics in SLH at the I-ORS. This study confirmed that a novel stuck check using the reoccurrence rate of the same value for a specific period can reduce truncation and increase the retention rate of good data compared to existing QC processes such as IOC.





To evaluate the reliability of SLH data applying the TALOD and analyze the characteristics of SLH data from
various institutions, we collected and compared with HYCOM, Satellite, GLORYS, and ORAS5. Before 2018,
HYCOMa and Satellite data exhibited the highest performance, while GLORYS and ORAS5 showed relatively
higher RMSE. Since 2018, the trend of SLH for HYCOM (HYCOMb) was –23.86 mm/yr, which showed
unrealistic results compared to other datasets. In conclusion, the reanalysis data, including HYCOMa and satellite
altimetry, showed a more similar pattern to the observation, and the others exhibited a quite narrower distribution
for anomalies. Through assessment, we confirmed an issue with the variability of SLH in HYCOM, and the
reliability and validity of the TALOD QC method and SLH observation at I-ORS.
The TALOD QC process includes the extreme event flag (EEF), which indicates the period during which SLH is
affected by extreme weather. For instance, since the variance of SLH was more than four times larger (including
flagged data) than usual during the typhoon-influenced period, some good data can be flagged as range and spike
errors. Ensuring sufficient observation numbers is crucial for research on typhoons. Therefore, we provide the
extreme event option so researchers can use these data for extreme weather dynamics.
In the budget analysis, the BS effect related to mass exchange between the ocean and land contributed significantly,
accounting for approximately 70% of the total sea level change. The difference in sea level trend between the I-
ORS and satellite altimetry (about 2.67 mm/yr) was attributed to VLM. The total VLM estimated from reanalysis
data (-2.17 mm/yr) indicates considerable ground subsidence at the I-ORS site. In detail, this subsidence was more
influenced by local processes than natural processes such as BS or GIA. Although the total VLM varies depending
on the reanalysis data, the GNSS-measured vertical displacement trend from 2013 to 2019 was calculated at -
0.89±0.47 mm/yr, demonstrating the ongoing ground subsidence at the I-ORS.
Despite the advancements in the TALOD QC process, several challenges remain. The TALOD QC process only
targets the observed SLH and is still not fully automated. Additionally, there is a need for further processes that
make it possible to take count of misclassification in extreme weather, such as rogue waves. In normal cases, good
data with extreme values induced by the inverted barometer and steric effect may be erroneously identified as
errors. Thus, an additional step of adjusting coefficients using atmospheric and oceanographic observation
variables is required.
Nevertheless, the TALOD QC process has the utility of being applied to both tide gauges and range finders. It
also utilizes predicted tidal components for each point, enhancing its adaptability. Well-controlled in-situ data are
essential not only for data assimilation and validation but also for data management. The I-ORS platform stands
out as a unique resource, offering over 20 years of continuous atmospheric and oceanographic observation data



in the open sea. Additionally, the Gageocho Ocean Research Station (G-ORS) and Socheongcho Ocean Research
Station (S-ORS) are positioned along the meridian, contributing to the study of marine environmental
development.
**Acknowledgement**
This research was supported by Korea Institute of Marine Science & Technology Promotion (KIMST) funded by
the Ministry of Oceans and Fisheries (RS-2021-KS211502 and RS-2022-KS221544 ), and by the Korea Institute
of Ocean Science & Technology (PEA0201).
**Author contributions**
Taek-bum Jeong: Conceptualization, Methodology, Formal analysis, Writing – original draft, Writing – review &
editing. Yong Sun Kim: Conceptualization, Methodology, Validation, Writing – review & editing, second contact
author. Hyeosoo Cha: Conceptualization, Methodology, Validation, Writing – review & editing. Kwang-Young
Jeong: Conceptualization, Methodology, Writing – review & editing. Mi-Jin Jang: Conceptualization,
Methodology, Writing – review & editing. Jin-Yong Jeong: Conceptualization, Methodology, Writing – review
& editing. Jae-Ho Lee: Conceptualization, Methodology, Validation, Writing – review & editing, first contact
author.
**Competing interests**
The contact author has declared that none of the authors has any competing interests.
**Special issue statement**
This article is part of the special issue "Oceanography at coastal scales: modelling, coupling, observations, and
applications".
**Acknowledgements**
This research was supported by Korea Institute of Marine Science & Technology Promotion (KIMST) funded by
the Ministry of Oceans and Fisheries (RS-2021-KS211502 and RS-2022-KS221544), and by the Korea Institute
of Ocean Science & Technology (PEA0201).



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



**List of Table**







semicircles centered at the "OBS" marker mean the Root Mean Square Errors. The colors and markers
indicate each data (black circle: observation, green square: Satellite, light cyan hexagram: GLORYS,
purple pentagram: ORAS5, red diamond: HYCOM, light red upward-pointing triangle: HYCOM-R, light
red downward-pointing triangle: HYCOM-S).
Figure 12. Monthly time series of sea level anomalies (left) and bar chart with error bar for sea level rise rate
(right; units: mm/yr). Each color and type of line indicates the dataset (OBS: black solid line, Satellite:
green solid line, Sum: bright red solid line, STERO: orange diamond, BARY: purple dotted line, GIA:
sky-blue dotted line, and GNSS: bright brown).
