# Peer review of "Application of quality-controlled sea level height observation"

_EGUsphere, 2024_

## Author Response (AR1)

RC1 Detailed comments:

1. Please do an analysis on the tides.

A: Harmonic analysis was conducted on the SLH observations during the well-observed period from March to June 2021. The M2 tide exhibits the largest amplitude of 0.62 m, with a signal-to-noise ratio (SNR) exceeding $10^3$. This tide is followed by S2 (0.32 m), K1 (0.20 m), N2 (0.16 m), and O1 (0.15 m). The mean amplitude of these primary constituents was 0.28 m, with an average SNR of approximately 3,000, notably higher than that of the remaining 31 constituents with amplitudes under 0.1 m (mean amplitude: 0.01 m, mean SNR: 6.01). We have reflected this tidal analysis result in the revised version [L215-218].

> [L215-218]
>
> Harmonic analysis of the observed SLH at the I-ORS shows that the M2 tide has the largest amplitude of 0.62 m. It is followed by S2 (0.32 m), K1 (0.20 m), N2 (0.16 m), and O1 (0.15 m). The mean amplitude of these primary constituents is 0.28 m, which is notably higher than that of the remaining 31 constituents with amplitudes under 0.1 m.

2. Please specify the constituents considered in this study.

A: We have specified the 15 tidal components for this study —M2, S2, N2, K2, 2N2, K1, O1, P1, Q1, Mf, Mm, M4, MN4, MS4, and S1— in the revised manuscript [L213-214].

> [L213-214]
>
> The monthly tidal data, consisting of 15 constituents (M2, S2, N2, K2, 2N2, K1, O1, P1, Q1, Mf, Mm, M4, MN4, MS4, and S1), were extracted from the TPXO9 and adjusted using the observed SLH for the same period (Fig. 4).

3. Fig. 1: characters should be larger.

A: Acknowledged. We have revised the font size in Figure 1 of the revised manuscript.

[Figure]

Figure 1. The structure of I-ORS and Instruments (Right) and the horizontal distribution for bathymetry and the tracks of typhoons passed by I-ORS (data from Joint Typhoon Warning Center; cases depicted in Fig. 6). The star marks indicate the location of the I-ORS (red) and the Socheongcho (black, north) and Gageocho (black, south) Ocean Research Stations. The black dots depict the locations of tide stations. The grey solid lines show the storm tracks passing by I-ORS from 2003 to 2022 (Table 2). The darker lines indicate the typhoon case in Fig. 6.

4. L250-253: There really should not be so many stuck errors in a modern sensor, is there an explanation for that?

A: Initially, we attempted to solve this issue by contacting the MIROS company, the manufacturer of the range finder. Then, they requested access to all related raw data to look into the cause of this unrealistic error. Unfortunately, we are not able to provide this access due to contractual restrictions. The technicians working at the Korea Hydrographic and Oceanographic Agency (KHOA), in charge of managing the ocean research station, suggested that the unstable power supply might cause the errors. In addition, we cannot rule out the possibility that biological floats contribute, at least in part, to this phenomenon, as their increase of stuck errors was observed during warm seasons, specifically spring (16,536), summer (7,985), autumn (3,067), and winter (5,795). This seasonality may indicate an impact of surface-drifting plankton (or material) on the rangefinder's reflection rate, presumably resulting in these recurrent stuck errors. To assess this hypothesis, we require an in-depth study on the relationship between floating materials and their impact on sea level observations with taking into account the electrical power level of this station. We have added the relevant content to the revised manuscript [L

284-289].

[L284-289]

Seasonal patterns in the frequency of each flag were further analyzed. The number of bad data occurrences was highest in spring, exceeding the annual average by a factor of 1.28. This seasonal increase was primarily driven by the higher incidence rate of stuck errors. Specifically, a total of 33,383 stuck errors were recorded, of which 16,536 occurred in spring—the highest among all seasons (winter: 5,795; summer: 7,985; autumn: 3,067). The frequency of stuck errors in spring was approximately twice the annual average, presumably reflecting the influence of surface-drifting plankton on the rangefinder's reflection rate during the spring bloom period.

5. L270: Yes, recurrent spikes make the automatic detection "think" they are good values. Good job, there, solving the issue by computing a local bias.

A: Thank you for your comment. These recurrent spikes and stuck values can be falsely classified as real, good data by a classical quality control process. The TALOD approach effectively addresses this issue by calculating non-tidal residuals in order to distinguish real observations from outliers. This improvement reduces false detections and enhances the reliability of observed sea levels.

6. L420 -HYCOM shows a trend in SLH of -23.86mm/yr, which is quite high. Please consider whether it is a good model to compare to.

A: HYCOM is a reanalysis dataset that assimilate various observational data, including temperature, salinity, and sea level height, into the NCODA system. This reanalysis data has been widely used for both initial and boundary conditions of numerical simulations, as well as for assessing model performance in the broad fields of oceanography and climate studies. When calculating a linear trend, the HYCOM reanalysis rendered an unrealistically high negative trend of -23.86 mm/yr; this impractical trend is primarily due to the robust sea level falling in the HYCOM's non-assimilation simulation during the recent period since 2017 [[L158-L160]. Therefore, in this study, we divided the 2003–2022 period into an assimilation-applied phase (HYCOM-R) and a non-assimilated phase (HYCOM-S) to discuss relevant issues in HYCOM's performance [L366-L370].

Our sea level analysis may reflect that the recent data of the HYCOM product is unsuitable for studying circulation and associated water properties, at least, in the East China Sea, although in-depth research is needed to assess the comprehensive skills of HYCOM compared to other reanalysis datasets (i.e., GLORYS12, BRAN2020, and ORAS5).

7. Sec 3.3 Great analysis of the contribution to sea level rise on this site.

A: We appreciate the compliment. Sea level rise is a critical topic linked to climate change. This study aimed to comprehensively analyse not only the sea level trend observed from fixed platforms but also the main contributions to sea level rise, including barostatic and steric effects, as well as vertical land motion. We hope that these findings will contribute to a deeper understanding of regional sea level changes and their implications for climate research.

RC2 Detailed comments:

1. Line 75 – Open ocean tides are generally easier to analyze than those at the coast, where shallow water effects can distort the tide.

A: We may have unintentionally given the impression that our main challenge was related to tidal complexity. Our primary focus is on how to classify error-like outliers in observed sea level data, rather than on the tidal complexity. The I-ORS data obtained through the rangefinder contains a considerable amount of unrealistic values, including overshooting-like errors, spikes, and a new form of stuck values, unlike coastal tide gauges that generally provide continuous and reliable measurements. These many error-like values cause frequent disruptions in the time series, making it difficult to extract consistent tidal components through harmonic analysis. This study aims to develop a quality control framework suitable for the error characteristics of range finder data in the open ocean, thereby preserving as much qualified data as possible. We have added the corresponding content to the revised manuscript [L87-94].

[L87-94]

They used the estimated tidal components to get residual components of SLH data and then performed outlier detection. Recently, Lin-Ye et al. (2023) expanded the existing SEa LEvel NEar-real-time (SELENE) QC software by incorporating additional modules to enable delayed-mode QC. In particular, the harmonic analysis-based de-tiding module was upgraded to remove tidal components. The resulting time series has been effectively utilized to identify subtle anomalies such as spikes, attenuation, and datum shifts by eliminating the periodic tidal variability from the original observations. This harmonic analysis-based approach is appropriate for the data stably obtained from tide gauge stations but seems impertinent to measurements in the open ocean, which may have various types of intricate outliers.

2. Line 152 S 2.2.1 Meta Check – This terminology is confusing. The term 'metadata' seems to be used in a non-traditional way here. It seems that what the authors are actually describing are cross checks between instrumental maintenance records and sea level time series. I would therefore give this check an alternative name.

A: Agreed. The term "Meta-Check" can be confusing to readers; hence, we renamed it "Manual Check" in the revised manuscript [L172].

3. Lines 156-163 are confusing, There is stated to be no maintenance record for the station, but then it

is claimed that the sensor was relocated twice and swapped out on another occasion. How did the authors deduce this in the absence of maintenance records?

A: You are right. There were no formal records detailing the relocations, configuration changes, or cleansing for this rangefinder sensor during most periods of operation, except for the recent few years. We collected maintenance information through personal discussions with technicians from KHOA and the commissioned company, responsible for managing this station. They reported two critical instances of sensor maintenance: first, a change in the data recording method on 12 December 2007, and later a sensor replacement due to its malfunction in 2016. These changes were confirmed with the recorded data. We have described this information in the revised manuscript [L176-181].

[L176-181]

This examination should be based on historical metadata information (or field notes) on the sensor's maintenance, cleansing, power shortage events of the station, etc. Unfortunately, metadata information concerning the observed SLH time series from the I-ORS was not made publicly available as documentation. Instead, considering the following processes, we flagged subjectively sections where the periodicity of the SLH data was irregular or nonsensical data existed for several days.

4. Line 168 – S 2.2.2 Stuck Check. It is unclear whether this check is performed manually or is automated. In any event, given that step 1 is a manual check, why would these 'stuck data' checks not be identified during step 1? Figure 5 d shows that they are quite obvious.

A: The stuck check in TALOD is an automatic process to particularly to detect a new form of stuck values. The manual check in step 1 was designed to ensure the accuracy of residual component estimation by removing non-realistic, error-like patterns that last over a day. This process manually flags only the periods that need to be removed for the next steps in the QC procedure. When attempting to detect stuck errors in Figure 5d by adopting manual or typical qc process, either the entire dataset from May 5 to 6 is flagged as errors or fails to detect this unique type of error, thus tending to classify them as good data (KHOA and SELENE methods in Figure A). Meanwhile, the newly designed stuck check, an automatic process in TALOD QC, allows us to retain most observed data successfully by flagging these unique stuck values. Figure A is provided as Figure S3 in the supplement.

[Figure]

Figure A. Same as Fig. 5, but for invariant stuck case (a-c, from 05 May 2005 to 07 May 2005), stuck case during short-period (d-f, from 12 Jul 2013 to 18 Jul 2013), and range-spike misclassification case (g-i, 12 Jun 2016 to 14 Jun 2016). The figures represent TALOD, KHOA, and SELENE results, respectively. The SELENE results were performed using a spike threshold (sigma) of 1.0, a window size of 36, a polynomial fitting degree of 3, a valid data range of –2.0 to 2.0, and a maximum of 30 iterations.

5. Line 177 – S 2.2.3 Range Check. I don't understand why the authors would go to the trouble of using predictions from a global tidal model to identify the tidal range within a given month, to then remove an offset to move the model closer to observations and then smooth the model tide to compare it to observations. Surely it would be far simpler to perform Classical Harmonic Analysis of the tide gauge observations to generate a non-tidal residual time series in which any suspect datapoints will be immediately obvious, because they are not masked by the dominant tidal variability? This is the principle on which conventional QC of sea level time series is built and by omitting this step, the authors are making life much more difficult for themselves and may indeed overlook some suspect data.

A: This study aims to develop a quality control procedure that is both applicable to data obtained from the I-ORS' rangefinder, which observes sea level with a substantial amount of error-like data, and generalizable to observations from a typical tidal gauge in the coastal region. Due to the limited length, coverage, or poor quality of observations such as those in this study, many researchers struggle to obtain accurate tidal components through a typical harmonic analysis. Practically speaking, we initially adopted the least squares method and harmonic analysis to process our observations at an early stage; this conventional approach, however, did not yield adequate residuals for a local range check (see the attached supplementary Figures A and B, as well as the response provided below for a detailed discussion).

To address this limitation, we took advantage of a well-validated tide prediction model. Some may point out the drawback of restricting the standalone operation of this approach, as it relies on the output of an external tidal model, as well as a script for extracting tides from the model. However, the outputs and processing scripts of various (or localized) tidal models are publicly provided and easily accessible, for instance, from https://www.tpxo.net/global of Oregon State University. We believe that using tidal models may be a practical alternative to the conventional method, as in our specific situation, where power supply issues accompany large stuck and spike errors.

It is noteworthy to point out that this study uses a double smoothing technique to improve the performance of the spike check that follows the range check. This approach allows us to set thresholds more narrowly to 0.5 meters and considerably increases the spike-detection rate by reducing misclassifications caused by frequent overshooting. Figure B is provided as Figure S1 in the supplement.

[Figure]

Figure B. Time series of non-tidal residual values for cases involving about 4 flags. a) manual, b) range, c) spike, and d) stuck. Each marker indicates good data (grey circle), manual (blue circle), range (green triangle), spike (yellow square with red outline), and stuck (red cross), respectively.

6. Figure 5(b) and (c) It isn't clear to me that the range or spike check has worked as some of the yellow boxes appear visually to be within range. If they are truly out-of-range, that will be apparent in the non-tidal residual time series, which should be presented in Figure 5 instead of the total water level.

A: The range check in TALOD is performed on residuals after removing tides; the spike check is also performed based on the square of the change rate of those residuals. This discrepancy appears to invoke confusion about whether the quality control process is being conducted correctly. We checked the non-tidal residual time series and confirmed that the range and spike checks were functioning properly. In

response to your comment, we have added the non-tidal residual time series (Figure A) to the supplementary material as Figure S3.

7. Figure 6 I'm not clear on the purpose of the EEF flag. Are the authors trying to remove real variability that is due to typhoons? Some of those datapoints that are flagged look reasonable but whether or not this is truly the case can only be demonstrated in a non-tidal residual time series.

A: The purpose of the EEF flag is not to remove data but to provide users with information, thereby promoting its usability by informing them that these observations, even those marked with precedent qc flags for spikes and out-of-range observations, may still be good data. We manually assigned an extreme event flag to data observed during extreme weather periods. Then, the users have the right to choose the data for their scientific objectives. We have added the corresponding content to the revised manuscript [L251-252].

[L251-252]

As a final QC procedure, this study introduced the extreme event flag (EEF) to allow users with an option to utilize the data based on their scientific objectives.

We calculated the non-tidal residuals and confirmed that the outliers were flagged by the range and spike checks in response to the reviewer's comment (Figure C). Additionally, we have included the residual time series for Figure 6 in the supplementary material (Figure S2).

[Figure]

Figure C. Time series of non-tidal residual values for each typhoon case. a) Bolaven in 2012, b) Soulik in 2018, c) Lingling in 2019, and d) Bavi in 2020. Good data (grey circle), EEF (purple circle), range (green triangle), and spike (yellow square with red outline), respectively.

8. Line 235 the authors state that they have compared their process with the IOC standard methodology, but they do not provide a reference for the IOC QC regimen that has been used nor do they describe the software that was used to do so. Given that harmonic analysis is a fundamental component of the IOC QC methodology, I can't see that such a comparison is valid.

A: Appreciate this valuable comment. The Korea Hydrographic and Oceanographic Agency (KHOA) has conducted quality control on the observational data in accordance with the IOC manuals[1,2] and the NOAA handbook[3]. This study utilized the processed data provided by KHOA, which we refer to as IOC QC. The KHOA's methodology is for a near-real-time process after a real-time automatic QC process. They used only four major tidal components, which were estimated from long-term sea level data, to calculate tide residuals, then flag residual components that exceed thresholds. And we have added the relevant details in Section 3.1 [L273-275] and Table 4. To avoid confusion, we replaced 'IOC' with 'KHOA' in the revised version and included the corresponding references.

L567: [1]IOC, GTSPP Real-Time Quality Control Manual, 1990

L568: [2]IOC, Manual of Quality Control Procedures for Validation of Oceanographic Data, 1993

L620: [3]NOAA, NDBC Handbook of Automated Data Quality Control Checks and Procedures, 2009

9. Line 260 – Are the authors simply flagging real extreme events as bad?

A: No. As mentioned above, data observed during extreme events may exhibit large and abrupt variability, but be realistic in-situ observations. Therefore, we manually assigned the Extreme Event Flag (EEF) to provide users with an option to utilize the data based on their scientific objectives [L251-252].

> [L251-252]
>
> As a final QC procedure, this study introduced the extreme event flag (EEF) to allow users with an option to utilize the data based on their scientific objectives.

10. Line 273 mention automatic QC and it isn't clear to me whether the TALOD QC method is manual or automatic, nor whether the IOC QC protocol that has been used is an automatic or delayed mode process. I'm not sure that the 2 systems are comparable.

A: The TALOD method automates procedures except for the "Meta Check (manual check in the revised

version)" and the "Extreme Event Flag" as the last step. The KHOA QC process encompasses both manual and automatic processes. In this study, we focused on comparing and analyzing the results obtained by the automatic processes of both QC methods.

11. Figure 8 – the results reported to be from the IOC methodology do not look correct to me. I would recommend that the authors consult the IOC manual of QC Quality control of in situ sea level observations: a review and progress towards automated quality control, volume 1 - UNESCO Digital Library. A recent publication which might also help is here OS - Delayed-mode reprocessing of in situ sea level data for the Copernicus Marine Service.

A: As recommended by the reviewer, we adopted SELENE's quality control and tide-surge module to examine the results, with a particular focus on the first (Fig. 8a, 8b) and third (Fig. 8e, 8f) cases (see the attached Figure A). The first case was a stuck error in which the values NaN, 0.088, and 0.090 alternated repeatedly. Unlike the TALOD, the SELENE one did not detect such errors. The third case corresponds to the range and spike checks. When two or more overshooting values occurred consecutively, the SELENE tends to result in misclassification or detection failure. We confirmed that the SELENE performs well for most datasets, but a specific case, such as our observations, seems to require additional handling. Our TALOD may offer a complementary approach that could be valuable as open-sea observations continue to expand. The content related to SELENE is described in the Introduction [L88-92], Results [L319-323] sections, and the supplementary material is provided as Figure S3.

[Figure]

Figure A. Same as Fig. 5, but for invariant stuck case (a-c, from 05 May 2005 to 07 May 2005), stuck case during short-period (d-f, from 12 Jul 2013 to 18 Jul 2013), and range-spike misclassification case (g-i, 12 Jun 2016 to 14 Jun 2016). The figures represent the processed sea level heights by adopting

TALOD, KHOA, and SELENE QCs, respectively. The SELENE results were performed using a spike threshold of 1.0, a window size of 36, a polynomial fitting degree of 3, a valid data range of –2.0 to 2.0, and a maximum of 30 iterations.

12. Line 378-392 – The observed VLM at a tide gauge site from the GNSS receiver is a better indicator than differencing altimetry etc, but in any event the observed VLM from GNSS appear to act in the opposite sense to the one the authors have derived.

A: All three VML estimations do appear negative trends, i.e., obtained by subtracting the difference between satellite altimetry and observations -2.51 ± 0.62 mm/yr, by summing the VLM of processes -2.17 ± 0.89 mm/yr, and from using a GNSS sensor -0.89 ± 0.47 mm/yr. These trends consistently reflect the subsidence of the I-ORS' ground and do not act in the opposite sense.